# Mapping two-dimensional polar active fluids to two-dimensional soap and one-dimensional sandblasting

Leiming Chen[1], Chiu Fan Lee[2] & John Toner[3,4]

Active fluids and growing interfaces are two well-studied but very different non-equilibrium systems. Each exhibits non-equilibrium behaviour distinct from that of their equilibrium counterparts. Here we demonstrate a surprising connection between these two: the ordered phase of incompressible polar active fluids in two spatial dimensions without momentum conservation, and growing one-dimensional interfaces (that is, the 1+1-dimensional Kardar–Parisi–Zhang equation), in fact belong to the same universality class. This universality class also includes two equilibrium systems: two-dimensional smectic liquid crystals, and a peculiar kind of constrained two-dimensional ferromagnet. We use these connections to show that two-dimensional incompressible flocks are robust against fluctuations, and exhibit universal long-ranged, anisotropic spatio-temporal correlations of those fluctuations. We also thereby determine the exact values of the anisotropy exponent $\zeta$ and the roughness exponents $\chi_{x,y}$ that characterize these correlations.

[1] Department of Physics, College of Science, China University of Mining and Technology, Xuzhou, Jiangsu 221116, China. [2] Department of Bioengineering, Imperial College London, South Kensington Campus, London SW7 2AZ, UK. [3] Department of Physics and Institute of Theoretical Science, University of Oregon, Eugene, Oregon 97403, USA. [4] Max Planck Institute for the Physics of Complex Systems, Nöthnitzer Strasse 38, 01187 Dresden, Germany. Correspondence and requests for materials should be addressed to L.C. (email: leiming@cumt.edu.cn) or to C.F.L. (email: c.lee@imperial.ac.uk) or to J.T. (email: jjt@uoregon.edu).

Non-equilibrium systems can behave radically different from their equilibrium counterparts. Two of the most striking examples of such exotic non-equilibrium behaviour are moving interfaces (for example, the surface of a growing crystal)[1], and 'flocks' (that is, coherently moving states of polar active fluids)[2–7]. The former is described by the Kardar–Parisi–Zhang (KPZ) equation[8], which is also a model for erosion (that is, sandblasting). This equation predicts that a two-dimensional (2D) moving interface (that is, the surface of a three-dimensional crystal) is far rougher than the surface of a crystal in equilibrium. In contrast, hydrodynamic theories of polar active fluids[9–13] predict that a large collection of 'active' (that is, non-equilibrium) moving particles (which could be anything from motile organisms to molecular motor propelled biological macro-molecules[2–7,9–17]) can develop long-ranged orientational order in 2D, while their equilibrium counterparts (for example, ferromagnets), by the Mermin–Wagner[18,19] theorem, cannot. At the same time, many non-equilibrium systems can also be mapped onto equilibrium systems[20]; an example of this that proves very relevant is the connection between the $1+1$-dimensional KPZ model and the defect-free 2D smectic (that is, soap) model[21,22]. Here, we add a living system to this list by showing that generic incompressible polar active fluids, for example, an incompressible bird flock, also belongs to the same universality class.

Since many fluids flow much slower than the speed of sound, a great deal of the work done over the past two centuries on equilibrium fluids has focused on incompressible fluids[23,24]. In this paper, we consider 2D incompressible active fluids; more specifically, we consider them in rotation invariant, but non-Galilean-invariant situations in which momentum is not conserved (for example, active fluids moving over an isotropic frictional substrate such as cells crawling on a substrate). Such an active system contains rich physics: it has recently been shown that their static-moving transition belongs to a new universality class[25]. Here, we focus on the long-range properties of the system in the moving phase.

We note that the incompressibility condition is not merely a theoretical contrivance; not only can it be readily simulated[26,27] but it can arise in a variety of real experimental situations, including systems with long-ranged repulsive interactions[28], and dense systems of active particles with strong repulsive short-ranged interactions, such as bacteria[26]. In addition, incompressibility plays an important role in the motile colloidal systems in fluid-filled microfluidic channels recently studied[29], although these systems differ in detail from those we study here in being two-component (background fluid plus colloids).

In this paper, we formulate a hydrodynamic (that is, long-wavelength and long-time) theory of the ordered, moving phase of a 2D incompressible polar active fluid. We find that the equal-time velocity correlation functions of the type of incompressible polar active fluids we study here can be mapped exactly onto those of two equilibrium problems: a divergence-free 2D XY model (a peculiar type of ferromagnet different from ordinary ferromagnets, which are divergenceful) and a dislocation-free 2D smectic A liquid crystal[21,22,30–32], as well as onto the time dependent correlation functions of the non-equilibrium $1+1$-dimensional KPZ equation[8]. The mapping of the 2D smectic onto the $1+1$-dimensional KPZ equation was discovered by Golubovic and Wang[21,22]; the other two mappings are new (although 2D ferromagnets with 2D dipolar interactions, which are similar but not identical systems, have also been mapped onto 2D smectics[30]). This series of mappings is illustrated in Fig. 1.

Our results imply in particular that incompressible polar active fluids can develop long-ranged orientational order (by developing a non-zero mean velocity $\langle \mathbf{v} \rangle$) in two dimensions, just as found

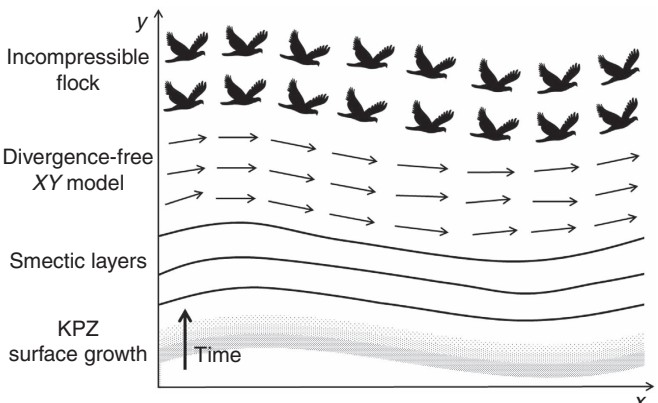

**Figure 1 | Visual representation of the mappings.** The flow lines of the ordered phase of a 2D incompressible polar active fluid, the magnetization lines of the ordered phase of divergence-free 2D XY magnets, dislocation-free 2D smectic layers and the surfaces of a growing one-dimensional crystal (which can be obtained by taking equal-time-interval snap shots) undulate in exactly the same way over space; their fluctuations share exactly the same asymptotic scaling behaviour at large-length scales. Note that the vertical axis is time for KPZ surface growth and the y Cartesian coordinate for the other three systems.

previously for compressible polar active fluids, but in complete contrast to ordinary divergenceful ferromagnets with underlying rotation invariance, which cannot so order. However, the scaling behaviour of the velocity correlation functions is very different from those for compressible polar active fluids studied in refs 11,12. Specifically, we find that the equal-time velocity correlation function in the ordered phase has the following limiting behaviours:

$$
\begin{aligned}
&\langle |\mathbf{v}(\mathbf{r}, t) - \mathbf{v}(\mathbf{r}', t)|^2 \rangle \\
&= \begin{cases} C_0 - A Y^{-2/3}, & \kappa \ll 1 \\ C_0 - \frac{9}{2} c^2 \frac{A}{X} e^{-\Phi(\kappa)} \left[ 1 + \frac{4}{9} \left( \frac{x-x'}{y-y'} \right)^2 \right], & \kappa \gg 1 \end{cases}
\end{aligned} \tag{1}
$$

where $X \equiv |x-x'|/\xi_x$ and $Y \equiv |y-y'|/\xi_y$ are rescaled lengths in the $x$ and $y$ directions, and we define the scaling ratio $\kappa \equiv \frac{X}{Y^{2/3}}$. Here the function $\Phi(\kappa \gg 1) \approx c\kappa^3$ and the constant $c \approx 0.295$ are both universal (that is, system-independent), while $C_0$ and $A$ are non-universal (that is, system-dependent), positive, finite constants, and $\xi_{x,y}$ are non-universal lengths. Note that the fact that $\langle |\mathbf{v}(\mathbf{r}, t) - \mathbf{v}(\mathbf{r}', t)|^2 \rangle$ goes to a finite value in the large separation limit $|\mathbf{r} - \mathbf{r}'| \to \infty$ implies long-ranged orientational order.

## Results

**Model.** We start with the hydrodynamic model for compressible polar active fluids without momentum conservation[9,11,12]:

$$
\partial_t \rho + \nabla \cdot (\mathbf{v}\rho) = 0 \tag{2}
$$

$$
\begin{aligned}
&\partial_t \mathbf{v} + \lambda_1 (\mathbf{v} \cdot \nabla)\mathbf{v} + \lambda_2 (\nabla \cdot \mathbf{v})\mathbf{v} + \lambda_3 \nabla(|\mathbf{v}|^2) \\
&= U\mathbf{v} - \nabla P - \mathbf{v}(\mathbf{v} \cdot \nabla P_2) + \mu_B \nabla(\nabla \cdot \mathbf{v}) + \mu_T \nabla^2 \mathbf{v} \\
&\quad + \mu_2 (\mathbf{v} \cdot \nabla)^2 \mathbf{v} + \mathbf{f}
\end{aligned} \tag{3}
$$

where $\mathbf{v}(\mathbf{r}, t)$, and $\rho(\mathbf{r}, t)$ are, respectively, the coarse grained continuous velocity and density fields. All of the parameters $\lambda_i (i = 1 \to 3)$, $U$, the 'damping coefficients' $\mu_{B,T,2}$, the 'isotropic pressure' $P(\rho, v)$ and the 'anisotropic pressure' $P_2(\rho, v)$ are, in general, functions of the density $\rho$ and the magnitude $v \equiv |\mathbf{v}|$ of the local velocity. Note that we omit higher order damping terms

because, as our analysis will show later, they are irrelevant. In addition, because we focus here on the ordered phase, $\mu_{\mathrm{T,B,2}}$ is taken to be positive, as required for the stability of the ordered phase.

The $U$ term makes the local $\mathbf{v}$ have a non-zero magnitude $v_0$ in the ordered phase, by the simple expedient of having $U > 0$ for $v < v_0$, $U = 0$ for $v = v_0$, and $U < 0$ for $v > v_0$. The $\mathbf{f}$ term is a random driving force. It is assumed to be Gaussian with white noise correlations:

$$\langle f_i(\mathbf{r}, t) f_j(\mathbf{r}', t') \rangle = 2 D \delta_{ij} \delta^d(\mathbf{r} - \mathbf{r}') \delta(t - t') \quad (4)$$

where the 'noise strength' $D$ is a constant parameter of the system, and $i$, $j$ denote Cartesian components. Note that in contrast to thermal fluids (for example, model A in ref. 24), we are concerned with active systems that are not momentum conserving. As a result, the leading contribution to the noise correlations is of the form depicted in equation (4).

We now take the incompressible limit by taking only the isotropic pressure $P$ to be extremely sensitive to departures from the mean density $\rho_0$. One could alternatively consider making $U(\rho, v)$ and $P_2(\rho, v)$ extremely sensitive to changes in $\rho$ as well. This would be appropriate for an active fluid near its 'active jamming'[33] transition, since in that regime a small change in the local density can change the speed from a non-zero value for $\rho < \rho_{\mathrm{jam}}$ to zero for $\rho > \rho_{\mathrm{jam}}$. We will discuss this case in a future publication.

Focusing here on the case in which only the isotropic pressure $P$ becomes extremely sensitive to changes in the density, we see that, in this limit, in which the isotropic pressure suppresses density fluctuations extremely effectively, changes in the density are too small to affect $U(\rho, v)$, $\lambda_{1,2,3}(\rho, v)$, $\mu_{\mathrm{B,T,2}}(\rho, v)$ and $P_2(\rho, v)$. As a result, in the incompressible limit taken this way, all of them effectively become functions only of the speed $v$; their $\rho$-dependence drops out since $\rho$ is essentially constant.

Another consequence of the suppression of density fluctuations by the isotropic pressure $P$ is that the continuity equation (2) reduces to the familiar condition for incompressible flow,

$$\nabla \cdot \mathbf{v} = 0, \quad (5)$$

which can, as in simple fluid mechanics, be used to determine the isotropic pressure $P$.

All of the above discussion taken together leads to the following equation of motion in tensor notation for an incompressible polar active fluid, ignoring irrelevant terms:

$$\partial_t v_m = -\partial_m P + U(v) v_m - \lambda_1(v) v_n (\partial_n v_m) - \lambda_4(v) v_m v_n v_\ell (\partial_n v_\ell)$$
$$+ \mu_{\mathrm{T}}(v) \partial_n \partial_n v_m + \mu_2(v) v_\ell v_n \partial_\ell \partial_n v_m + f_m,$$
$$(6)$$

where $\lambda_4(v) \equiv \frac{1}{v} \frac{dP_2(v)}{dv}$, and the $\lambda_2$ and $\mu_B$ terms vanish due to the incompressibility constraint $\nabla \cdot \mathbf{v} = 0$ on $\mathbf{v}$. In writing (6), we absorb a term $W(v)$ into the pressure $P$, where $W(v)$ is derived from $\lambda_3(v)$ by solving $\frac{1}{2v} \frac{dW}{dv} = \lambda_3(v)$.

We now analyse the implications of equation (6) for the ordered state.

**Linear theory.** In the ordered phase, the system spontaneously breaks rotational symmetry by moving on average along some spontaneously chosen direction which we call $\hat{x}$; we call the direction orthogonal to this $\hat{y}$. In the absence of fluctuations (that is, if we set the noise $\mathbf{f}$ in (6) to zero), the velocity will be the same everywhere in space and time, and have magnitude $v_0$, which we remind the reader is defined by $U(v_0) = 0$. We treat fluctuations by expanding $\mathbf{v}$ around $v_0 \hat{x}$, defining $\mathbf{u}(\mathbf{r}, t)$ as the

small fluctuation in the velocity field about this mean:

$$\mathbf{v} = (v_0 + u_x(\mathbf{r}, t)) \hat{x} + u_y(\mathbf{r}, t) \hat{y}. \quad (7)$$

Plugging equation (7) into equation (6) and expanding to linear order in $\mathbf{u}$, leads to a linear stochastic partial differential equation with constant coefficients. Like all such equations, this can be solved simply by spatio-temporally Fourier transforming, and solving the resultant linear algebraic equations for the Fourier transformed field $\mathbf{u}(\mathbf{q}, \omega)$ in terms of the Fourier transformed noise $\mathbf{f}(\mathbf{q}, \omega)$. We can thereby relate the two point correlation function $\langle |u_y(\mathbf{q}, \omega)|^2 \rangle$ to the known correlations (4) of the random force $\mathbf{f}$. Integrating the result over all frequencies $\omega$, and dividing by $2\pi$, gives the equal time, spatially Fourier transformed velocity autocorrelation $\langle |u_y(\mathbf{q}, t)|^2 \rangle$. Details of this straightforward calculation are given in 'Methods' section; the result is

$$\left\langle |u_y(\mathbf{q}, t)|^2 \right\rangle = \frac{D q_x^2}{2\alpha q_y^2 + \Gamma(\mathbf{q}) q^2} \approx \frac{D q_x^2}{2\alpha q_y^2 + \mu q_x^4}, \quad (8)$$

where $\Gamma(\mathbf{q}) \equiv \mu q_x^2 + \mu_{\mathrm{T}}^0 q_y^2$ with $\mu \equiv \mu_{\mathrm{T}}^0 + \mu_2^0 v_0^2$, where $\mu_{\mathrm{T,2}}^0$ are $\mu_{\mathrm{T,2}}(v)$ evaluated at $v = v_0$, and the second, approximate equality applies for all $\mathbf{q} \to 0$. This can be seen by noting that, for $q_y \gg q_x^2$ and $\mathbf{q} \to 0$, $q_y^2 \gg \Gamma(\mathbf{q}) q^2$, while for $q_y \lesssim q_x^2$ and $\mathbf{q} \to 0$, $\Gamma(\mathbf{q}) q^2 \approx \mu q_x^4$. Hence, in both cases, (which together cover all possible ranges of $\mathbf{q}$ for $\mathbf{q} \to 0$), the approximation $2\alpha q_y^2 + \Gamma(\mathbf{q}) q^2 \approx 2\alpha q_y^2 + \mu q_x^4$ is valid.

We can now obtain the real space transverse fluctuations

$$\left\langle u_y^2(\mathbf{r}, t) \right\rangle = \int_{q_x \gtrsim \frac{1}{L}} \frac{d^2 q}{(2\pi)^2} \left\langle |u_y(\mathbf{q}, t)|^2 \right\rangle, \quad (9)$$

where $L$ is the lateral extent of the system in the $x$ direction (its extent in the $y$ direction is taken for the purposes of this argument to be infinite). Note that the longitudinal fluctuations $\langle u_x^2(\mathbf{r}, t) \rangle$ are negligible compared with $\langle u_y^2(\mathbf{r}, t) \rangle$. Using (8), the integral in (9) is readily seen to converge in the infra-red, and, hence, as system size $L \to \infty$. Since the integral is finite, and proportional to the noise strength $D$, it is clear that, for sufficiently small $D$, the transverse fluctuations $\langle u_y^2(\mathbf{r}, t) \rangle$ can be made small enough that long-ranged orientational order—that is, a non-zero $\langle \mathbf{v}(\mathbf{r}, t) \rangle$—is preserved in the presence of fluctuations; therefore, the ordered state is stable against fluctuations for sufficiently small noise strength $D$.

We show in the next section that this conclusion remains valid when nonlinear effects are taken into account (even though those non-linearities change the scaling laws from those predicted by the linear theory).

**Nonlinear Theory.** We begin by expanding the full equation of motion (6) to higher order in $\mathbf{u}$. This gives

$$\partial_t u_m = -\partial_m P - 2\alpha u_x \delta_{mx} - \lambda_1^0 v_0 \partial_x u_m + \mu_{\mathrm{T}}^0 \nabla^2 u_m + \mu_2^0 v_0^2 \partial_x^2 u_m + f_m$$
$$- \frac{\alpha}{v_0} \left( \frac{u_y^3}{v_0} \delta_{my} + 2 u_x u_y \delta_{my} + u_y^2 \delta_{mx} \right) - \lambda_1^0 u_y \partial_y u_y \delta_{my},$$
$$(10)$$

where the superscript '0' means that the $v$-dependent coefficients are evaluated at $v = v_0$, and we define the 'longitudinal mass' $\alpha \equiv -\frac{v_0}{2} \left( \frac{dU(v)}{dv} \right)_{v = v_0}$.

The first line of equation (10) contains the linear terms, including the noise $\mathbf{f}$; the first three terms on the second line are the relevant non-linearities, while the fourth term proves to be irrelevant, as we will soon show.

In writing (10), we have neglected 'obviously irrelevant' terms, by which we mean terms that differ from those explicitly

displayed in (10) by having more powers of the small fluctuations **u**, or more spatial derivatives of a given type. For more discussion of these 'obviously irrelevant' terms, see 'Methods' section. Note that only one of the non-linearities associated with the $\lambda_{1,2,3}$ terms, namely, $\lambda_1^0 u_y \partial_y u_y \delta_{my}$ actually remains at this point.

To proceed further, we must power count more carefully.

We only need to calculate one of the two fields $u_{x,y}$, since they are related by the incompressibility condition $\nabla \cdot \mathbf{v} = 0$. We choose to solve for $u_y$; its Fourier transformed equation of motion can be obtained by Fourier transforming (10) and acting on both sides of the resultant equation with the transverse projection operator $P_{lm}(\mathbf{q}) = \delta_{lm} - q_l q_m / q^2$ which projects orthogonal to the spatial wavevector $\mathbf{q}$. This eliminates the pressure term. Taking the $l = y$ component of the resulting equation gives (neglecting higher order gradient terms):

$$\partial_t u_y(\mathbf{q}, t) = -i v_1 q_x u_y(\mathbf{q}, t)$$

$$- \Gamma(\mathbf{q}) u_y(\mathbf{q}, t) + P_{yx}(\mathbf{q}) \mathscr{F}_{\mathbf{q}} \left[ -2\alpha \left( u_x(\mathbf{r}, t) + \frac{u_y^2(\mathbf{r}, t)}{2v_0} \right) \right]$$

$$+ P_{yy}(\mathbf{q}) \mathscr{F}_{\mathbf{q}} \left[ -\frac{\alpha}{v_0} \left( \frac{u_y^3}{v_0} + 2 u_x u_y \right) - \lambda_1^0 u_y \partial_y u_y \right] + P_{ym}(\mathbf{q}) f_m(\mathbf{q}, t),$$

(11)

where $\mathscr{F}_{\mathbf{q}}$ represents the Fourier component at wavevector $\mathbf{q}$, that is, $\mathscr{F}_{\mathbf{q}}[g(\mathbf{r})] \equiv \int d^2 r\, g(\mathbf{r}) e^{-i\mathbf{q}\cdot\mathbf{r}}$; the 'bare' value of the speed $v_1$, before rescaling and renormalization, is $v_1 = \lambda_1^0 v_0$, and $\Gamma(\mathbf{q})$ is given after equation (8).

We now rescale co-ordinates $(x, y)$, time $t$ and the components of the real space velocity field $u_{x,y}(\mathbf{r}, t)$ according to

$$x \mapsto e^{\ell} x, \quad y \mapsto e^{\zeta \ell} y, \quad t \mapsto e^{z\ell} t$$ (12)

$$u_y(\mathbf{r}, t) \mapsto e^{\chi_y \ell} u_y(\mathbf{r}, t),$$ (13)

$$u_x(\mathbf{r}, t) \mapsto e^{\chi_x \ell} u_x(\mathbf{r}, t) = e^{(\chi_y + 1 - \zeta)\ell} u_x(\mathbf{r}, t),$$ (14)

where the scalings of $u_x(\mathbf{r}, t)$ and $u_y(\mathbf{r}, t)$ are related by the incompressibility condition. Note that our convention for the anisotropy exponent here is exactly the opposite of that used in refs 9–13; that is, we define $\zeta$ by $q_y \sim q_x^\zeta$ being the dominant regime of wavevector, while refs 9–13 define this regime as $q_x \sim q_y^\zeta$.

Upon this rescaling, the form of equation (11) remains unchanged, but the various coefficients become dependent on the rescaling parameter $\ell$.

Details of this simple power counting (including the slightly subtle question of how to rescale the projection operators) are given in 'Methods' section. The results for the three parameters (damping coefficient $\mu$, 'longitudinal mass' $\alpha$ and noise strength $D$) that control the size of the fluctuations in the linear theory are: $\mu \mapsto e^{(z-2)\ell} \mu$, $\alpha \mapsto e^{(z-2\zeta+2)\ell} \alpha$, and $D \mapsto e^{(z-2\chi_y-\zeta-1)\ell} D$.

We now use the standard renormalization group logic to assess the importance of the non-linear terms in (11). This logic is to choose the rescaling exponents $z$, $\zeta$ and $\chi_y$ so as to keep the size of the fluctuations in the field **u** fixed on rescaling. This is clearly accomplished by keeping $\alpha$, $\mu$ and $D$ fixed. From the rescalings just found, this leads to three simple linear equations in the three unknown exponents $z$, $\zeta$ and $\chi_y$; solving these, we find the values of these exponents in the linearized theory: $\zeta_{\lin} = z_{\lin} = 2$, $\chi_{y\lin} = -1$. With these exponents in hand, we can now assess the importance of the non-linear terms in (11) at long-length scales, simply by looking at how their coefficients rescale. (We do not have to worry about the size of the actual non-linear terms themselves changing on rescaling, because we have chosen the rescalings to keep them constant in the linear theory.) We find that all of the non-linearities whose coefficients are proportional

to $\alpha$ are 'relevant' (that is, grow on rescaling), while those associated with the last remaining non-linearity, $\lambda_1^0$, associated with the $\lambda$ terms get smaller on rescaling: $\lambda_1^0 \mapsto e^{-\frac{\ell}{2}} \lambda_1^0$. Hence, this term will not affect the long-distance behaviour, and can be dropped from the problem. This is very different from the compressible problem, in which the $\alpha$ non-linearities are unimportant, while the $\lambda$ ones dominate; the reasons for this difference are discussed in 'Methods section'.

Dropping the $\lambda_1^0$ term in (10), and making a Galilean transformation to a 'pseudo-co-moving' co-ordinate system moving in the direction $\hat{x}$ of mean flock motion at speed $v_1 \equiv \lambda_1^0 v_0$ to eliminate the 'convective term' $v_1 \partial_x u_m$ from the right-hand side of (10), leaves us with our final simplified form for the equation of motion:

$$\partial_t u_m = -\partial_m P - 2\alpha \left( u_x + \frac{u_y^2}{2v_0} \right) \delta_{xm}$$

$$- \frac{2\alpha}{v_0} \left( u_x + \frac{u_y^2}{2v_0} \right) u_y \delta_{ym}$$

$$+ \mu \partial_x^2 u_m + \mu_T^0 \partial_y^2 u_m + f_m.$$

(15)

We now show that equation (15) also describes an equilibrium system: the ordered phase of the 2D $XY$ model subject to the divergence-free constraint $\nabla \cdot \mathbf{M} = 0$, where $\mathbf{M}$ is the magnetization. This connection enables us to use purely equilibrium statistical mechanics (in particular, the Boltzmann distribution) to determine the equal-time correlations of 2D incompressible polar active fluids.

**Divergence-free 2D XY model**. The 2D $XY$ model describes a 2D ferromagnet whose magnetization field $\mathbf{M}(\mathbf{s})$ and position $\mathbf{r}$ both have two components. The Hamiltonian for this model can be written, ignoring irrelevant terms, as[34]

$$H_{\mathrm{XY}} = \int d^2 r \left[ V(|\mathbf{M}|) + \frac{1}{2} \mu \left| \vec{\nabla} \mathbf{M} \right|^2 \right],$$ (16)

where $\mu$ is the 'spin-wave stiffness'. In the ordered phase, the 'potential' $V(|\mathbf{M}|)$ has a circle of global minima at a non-zero value of $|\mathbf{M}|$, which we will take to be $v_0$.

Expanding in small fluctuations about this minimum by writing $\mathbf{M} = (v_0 + u_x)\hat{x} + u_y \hat{y}$, we obtain, keeping only 'relevant' terms,

$$H_{\mathrm{XY}} = \frac{1}{2} \int d^2 r \left[ 2\alpha \left( u_x + \frac{u_y^2}{2v_0} \right)^2 + \mu \left| \vec{\nabla} \mathbf{u} \right|^2 \right],$$ (17)

where we define the 'longitudinal mass' $2\alpha \equiv \frac{\partial^2 V}{\partial |\mathbf{M}|^2} \big|_{|\mathbf{M}| = v_0}$.

We now add to this model the divergence-free constraint $\nabla \cdot \mathbf{M} = 0$, which obviously implies $\nabla \cdot \mathbf{u} = 0$. To enforce this constraint, we introduce to the Hamiltonian a Lagrange multiplier $P(\mathbf{r})$:

$$H' = H_{\mathrm{XY}} - \int d^2 r\, P(\mathbf{r})(\nabla \cdot \mathbf{u}).$$ (18)

The simplest dynamical model that relaxes back to the equilibrium Boltzmann distribution $e^{-\beta H'(\mathbf{u})}$ for the Hamiltonian $H'$ is the time-dependent-Ginsburg–Landau model[34,35] $\partial_t u_l = -\delta H'/\delta u_l + f_l$, where **f** is the thermal noise whose statistics can also be described by equation (4) with $D = k_B T = 1/\beta$. This time-dependent-Ginsburg–Landau equation is readily seen to be exactly equation (15) with $\mu_T^0 = \mu$. Therefore, we conclude that the ordered phase of 2D incompressible polar active fluids has the same static (that is, equal-time) scaling behaviours as the ordered phase of the 2D $XY$ model subject to the constraint $\nabla \cdot \mathbf{M} = 0$.

This mapping between a non-equilibrium active fluid model and a 'divergence-free' $XY$ model allows us to investigate the

fluctuations in our original active fluid model by studying the partition function of the equilibrium model.

To deal with the exact identity $\nabla \cdot \mathbf{u} = 0$, we use a trick familiar from the study of incompressible fluid mechanics: we introduce a 'streaming function'; that is, a new scalar field $h(\mathbf{r})$ such that

$$u_x = -v_0 \partial_y h, \quad u_y = v_0 \partial_x h. \quad (19)$$

Because this construction guarantees that the incompressibility condition $\nabla \cdot \mathbf{u} = 0$ is automatically satisfied, there is no constraint on the field $h(\mathbf{r})$.

The field $h(\mathbf{r})$ has a simple interpretation as the displacement of the fluid-flow lines from set of parallel lines along $\hat{x}$ that would occur in the absence of fluctuations, as illustrated in Fig. 2. (We thank Pawel Romanczuk for pointing out this pictorial interpretation to us.) This fact, which is explained in more detail in 'Methods' section, is a consequence of the fact that, as in conventional 2D fluid mechanics, contours of the streaming function are flow lines.

This picture of a set of lines that 'wants' to be parallel being displaced by a fluctuation $h(\mathbf{r})$ looks very much like a 2D smectic liquid crystal (that is, 'soap'), for which the layers are actually 1D fluid stripes.

**2D smectic and KPZ models**. This resemblance between our system and a 2D smectic is not purely visual. Indeed, making the substitution (19), the Hamiltonian (17) becomes (ignoring irrelevant terms like $(\partial_x \partial_y h)^2$, which is irrelevant compared with $(\partial_x^2 h)^2$ because $y$-derivatives are less relevant than $x$-derivatives):

$$H_s = \frac{1}{2} \int d^2 r \left[ B \left( \partial_y h - \frac{(\partial_x h)^2}{2} \right)^2 + K (\partial_x^2 h)^2 \right], \quad (20)$$

where $B = 2\alpha v_0^2$ and $K = \mu v_0^2$. This Hamiltonian is exactly the Hamiltonian for the dislocation-free 2D smectic model with $h(\mathbf{r})$ in equation (20) interpreted as the displacement field of the smectic layers, as also illustrated in Fig. 2.

The scaling behaviours of the dislocation-free 2D smectic model are extremely non-trivial, since the 'critical dimension' $d_c$ below which a purely harmonic description of these systems breaks down is $d_c = 3$ (ref. 36). Fortunately, these non-trivial scaling behaviours are known, thanks to an ingenious further mapping[21,22] of this problem onto the $1+1$-dimensional KPZ equation[8], which is a model for interface growth or erosion (for example, 'sandblasting'). In this mapping, which connects the equal-time correlation functions of the 2D smectic to the KPZ

equation, the $y$-coordinate in the smectic is mapped onto time $t$ in the KPZ equation with $h(x, t)$ the height of the 'surface' at position $x$ and time $t$ above some reference height. As a result, the dynamical exponent $z_{KPZ}$ of the $1+1$-dimensional KPZ equation becomes the anisotropy exponent $\zeta$ of the 2D smectic. Since the scaling laws of the $1+1$-dimensional KPZ equation are known exactly[8], those of the equal-time correlations of the 2D smectic can be obtained as well.

This gives $\zeta = 3/2$ and $\chi_h = 1/2$ as the exponents for the 2D smectic[21,22], where $\chi_h$ gives the scaling of the smectic layer displacement field $h(\mathbf{r})$ with spatial coordinate $x$. Given the streaming function relation (19) between $h(\mathbf{r})$ and $\mathbf{u}(\mathbf{r})$, we see that the scaling exponent $\chi_y$ for $u_y$ is just $\chi_y = \chi_h - 1 = -1/2$ and that the scaling exponent $\chi_x$ for $u_x$ is just $\chi_x = \chi_y + 1 - \zeta = -1$. Note that these exponents are different from those for compressible polar active fluids[9–13] where $\zeta = 5/3$ and $\chi_y = -1/5$. (Note that our convention here $(q_y \sim q_x^\zeta)$ is the inverse of that $(q_x \sim q_y^\zeta)$ used in refs 9–13.)

The fact that both of the scaling exponents $\chi_y$ and $\chi_x$ are less than zero implies that both $u_y$ and $u_x$ fluctuations remain finite as system size $L \to \infty$; this, in turn, implies that the system has long-ranged orientational order since $\langle |\mathbf{v}(\mathbf{r}, t) - \mathbf{v}(\mathbf{r}', t)|^2 \rangle$ remains finite as $|r - r'| \to \infty$. That is, the ordered state is stable against fluctuations, at least for sufficiently small noise $D$.

The velocity correlation function can be calculated through the connection between $\mathbf{u}$ and $h$. Using the aforementioned connection between 2D smectics and the $1+1$-dimensional KPZ equation, the equal-time layer displacement correlation function takes the form[21,22]:

$$C_h(\mathbf{r} - \mathbf{r}') \equiv \left\langle [h(\mathbf{r}, t) - h(\mathbf{r}', t)]^2 \right\rangle \quad (21)$$
$$= B|x - x'| \Psi(\kappa).$$

where we define the scaling variable $\kappa \equiv \frac{X}{Y^{2/3}}$, with $X \equiv |x - x'|/\xi_x$, and $Y \equiv |y - y'|/\xi_y$, and the non-universal constant $B$ is an overall multiplicative factor; estimates of the non-universal nonlinear lengths $\xi_{x,y}$ are given in 'Methods' section.

The limiting behaviours of the universal scaling function $\Psi$ have been studied numerically previously[37,38]. Here, we use the most accurate version currently known (www-m5.ma.tum.de/KPZ)[39,40]:

$$\Psi(\kappa) \approx \begin{cases} \Psi(\kappa) \approx c_1 + e^{-\Phi_h(\kappa)}, & \kappa \gg 1 \\ \kappa + \frac{c_2}{\kappa}, & \kappa \ll 1, \end{cases} \quad (22)$$

where for $\kappa \gg 1$,

$$\Phi_h(\kappa) = c\kappa^3 + \mathcal{O}(\kappa). \quad (23)$$

Here, the constants $c$ and $c_{1,2}$ are all universal and are given by $c \approx 0.295$, $c_1 \approx 1.843465$, and $c_2 \approx 1.060...$[39,40].

Rewriting the velocity correlation function (1) in terms of the fluctuation $\mathbf{u}$ using (7) gives

$$\left\langle |\mathbf{v}(\mathbf{r}, t) - \mathbf{v}(\mathbf{r}', t)|^2 \right\rangle \quad (24)$$
$$= C_0 - 2\langle u_y(\mathbf{r}, t) u_y(\mathbf{r}', t) \rangle - 2\langle u_x(\mathbf{r}, t) u_x(\mathbf{r}', t) \rangle,$$

where $C_0 = 2\langle u^2(\mathbf{r}, t) \rangle$ is finite, and the two correlation functions on the right-hand side of the equality are just the derivatives of the layer displacement correlation function:

$$\langle u_y(\mathbf{r}, t) u_y(\mathbf{r}', t) \rangle = -\frac{v_0^2}{2} \partial_x \partial_{x'} C_h(\mathbf{r} - \mathbf{r}'), \quad (25)$$

$$\langle u_x(\mathbf{r}, t) u_x(\mathbf{r}', t) \rangle = -\frac{v_0^2}{2} \partial_y \partial_{y'} C_h(\mathbf{r} - \mathbf{r}'). \quad (26)$$

To derive (25, 26) we use (19) and the definition of $C_h$ (that is, the first equality of formula (21)).

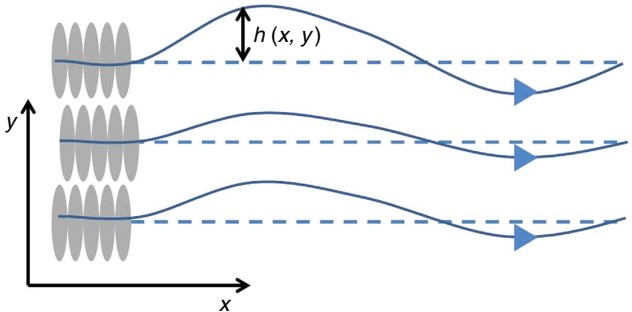

**Figure 2 | Analogy between displacements of flow lines and smectic layers.** In the case of 2D incompressible polar active fluids, the field $h(\mathbf{r})$ is the vertical displacement of the flow lines (that is, the solid lines) from the set of parallel lines (that is, the dotted lines) along $\hat{x}$ that would occur in the absence of fluctuations. For a defect-free 2D smectic, it likewise gives the vertical displacement of the smectic layers (that is, the solid lines) from their reference positions (that is, the dotted lines) at zero temperature.

Inserting (25, 26) into (24) and using the asymptotic forms (21, 22) for $C_h$, we obtain (as explained in more detail in 'Methods') the asymptotic form of the velocity correlation function given by (1). We can also obtain the Fourier transformed equal-time correlation functions; these are given in 'Methods' section.

## Discussion

We formulate a universal equation of motion describing the ordered phase of 2D incompressible polar active fluids. After using renormalization group analysis to identify the relevant non-linearities of this model, we perform a series of mathematical transformation which map our model to three other interesting, but seemingly unrelated, models. Specifically, we make heretofore unanticipated connections between four seemingly unrelated systems: the ordered phase of 2D incompressible polar active fluids, the ordered phase of the divergence-free 2D XY model, dislocation-free 2D smectics, and growing one-dimensional interfaces. Through this connection, we show that 2D incompressible polar active fluids spontaneously break continuous rotational invariance (which ordinary divergenceful ferromagnets cannot do), and obtain the exact scaling behaviour of the equal-time velocity correlation function of the original model. Because this mapping only involves equal-time correlations, the dynamical scaling of the original model is currently unknown. We hope to determine this scaling in further work.

## Methods

**Linear theory.** In this section, we give the details of the derivation of the linearized theory of incompressible polar active fluids. We begin with the linearized equation of motion, obtained by expanding equation (6) of the main text to linear order in the fluctuation $\mathbf{u}$ of the velocity around its mean value $v_0\hat{x}$:

$$\partial_t u_m = -\partial_m P - 2\alpha u_x \delta_{xm} - \lambda_1^0 v_0 (\partial_x u_m) - \lambda_4^0 v_0^3 \delta_{xm}(\partial_x u_x) \\ + \mu_T^0 \nabla^2 u_m + \mu_2^0 v_0^2 \partial_x^2 u_m + f_m, \quad (27)$$

where the superscript '0' means that the $v$-dependent coefficients are evaluated at $v = v_0$, and we define the 'longitudinal mass' $\alpha \equiv -\frac{v_0}{2}\left(\frac{dU(v)}{dv}\right)_{v=v_0}$.

Our goal now is to determine the scaling of the fluctuations $\mathbf{u}$ of the velocity with length and time scales, and to determine the relative scaling of the two Cartesian components $x$ and $y$ of position with each other, and with time $t$. That is, in the language of hydrodynamics, we seek the 'roughness exponents' $\chi_{x,y}$, the anisotropy exponent $\zeta$ and the dynamical exponent $z$ characterizing respectively the scaling of velocity fluctuations $u_{x,y}$, 'transverse' (that is, perpendicular to the direction of flock motion) position $y$ and time $t$ with 'longitudinal' (that is, parallel to the direction of flock motion) position $x$. Knowing this scaling (in particular, $\chi_{x,y}$) allows us to answer the most important question about this system: is the ordered state actually stable against fluctuations?

To obtain this scaling in the linear theory, we begin by calculating the fluctuations of $\mathbf{u}$ predicted by that theory. Since the two components of $\mathbf{u}$ are not independent, but, rather, locked to each other by the incompressibility condition $\nabla \cdot \mathbf{v} = 0$, it is only necessary to calculate one of them. We choose to focus on the $y$ component, which can be calculated by first spatio-temporally Fourier transforming (27), and then acting on both sides with the transverse projection operator $P_{lm}(\mathbf{q}) = \delta_{lm} - q_l q_m / q^2$ which projects orthogonal to the spatial wavevector $\mathbf{q}$. The component $\ell = y$ of the resultant equation then gives

$$-i(\omega - \lambda_1^0 v_0 q_x)u_y(\mathbf{q},\omega) = \left(2\alpha + i\lambda_4^0 v_0^3 q_x\right)\frac{q_x q_y}{q^2}u_x(\mathbf{q},\omega) - \Gamma(\mathbf{q})u_y(\mathbf{q},\omega) \\ + P_{ym}f_m(\mathbf{q},\omega), \quad (28)$$

where we define

$$\Gamma(\mathbf{q}) \equiv \mu_T^0 q^2 + \mu_2^0 v_0^2 q_x^2 = \mu q_x^2 + \mu_T^0 q_y^2, \quad (29)$$

with $\mu \equiv \mu_T^0 + \mu_2^0 v_0^2$.

We can eliminate $u_x$ from (28) using the incompressibility condition $\nabla \cdot \mathbf{v} = 0$, which implies, in Fourier space, $q_x u_x = -q_y u_y$. Solving the resultant linear algebraic equation for $u_y(\mathbf{q}, \omega)$ in terms of $f_m(\mathbf{q}, \omega)$ gives

$$u_y(\mathbf{q},\omega) = \frac{P_{ym}(\mathbf{q})f_m(\mathbf{q},\omega)}{-i[\omega - c(\hat{\mathbf{q}})q] + \Gamma(\mathbf{q}) + 2\alpha\left(\frac{q_y}{q}\right)^2}, \quad (30)$$

where we define the direction-dependent 'sound speed'

$$c(\hat{\mathbf{q}}) \equiv \lambda_1^0 v_0 \frac{q_x}{q} + \lambda_4^0 v_0^3 \frac{q_y^2 q_x}{q^3}. \quad (31)$$

Using equation (28), we can obtain $\langle |u_y(\mathbf{q},\omega)|^2 \rangle$ from the known correlations of the random force $\mathbf{f}$ (that is, formula (4) in the main text). Integrating the result over all frequencies $\omega$, and dividing by $2\pi$, gives the equal time, spatially Fourier transformed velocity autocorrelation:

$$\left\langle |u_y(\mathbf{q},t)|^2 \right\rangle = \frac{Dq_x^2}{2\alpha q_y^2 + \Gamma(\mathbf{q})q^2} \approx \frac{Dq_x^2}{2\alpha q_y^2 + \mu q_x^4}. \quad (32)$$

where the second, approximate equality applies for all $\mathbf{q} \to 0$. This can be seen by noting that, for $q_y \gg q_x^2$ and $\mathbf{q} \to 0$, $q_y^2 \gg \Gamma(\mathbf{q})q^2$, while for $q_y \lesssim q_x^2$ and $\mathbf{q} \to 0$, $\Gamma(\mathbf{q})q^2 \approx \mu q_x^4$. Hence, in both cases (which together cover all possible ranges of $\mathbf{q}$ for $\mathbf{q} \to 0$), the approximation $2\alpha q_y^2 + \Gamma(\mathbf{q})q^2 \approx 2\alpha q_y^2 + \mu q_x^4$ is valid.

Equation (32) implies that fluctuations diverge most rapidly as $\mathbf{q} \to 0$ if $\mathbf{q}$ is taken to zero along a locus in the $\mathbf{q}$ plane that obeys $q_y \lesssim q_x^2$; along such a locus, asymptotically, $\langle |u_y(\mathbf{q},t)|^2 \rangle \propto \frac{1}{q^2}$. In contrast, along all other locii, that is, those for which $q_y \gg q_x^2$, $\langle |u_y(\mathbf{q},t)|^2 \rangle \propto \frac{q_x^2}{q_y^2} \ll \frac{1}{q^2}$. In this sense, one can say that the regime $q_y \lesssim q_x^2$ shows the largest fluctuations at small $\mathbf{q}$; this implies the anisotropy exponent $\zeta = 2$.

We can get the dynamical exponent $z$ predicted by the linear theory by inspection of (30), although some care is required. The form of the first term in the denominator might suggest $\omega \propto q$, which would imply $z = 1$. However, the propagating $c(\hat{\mathbf{q}})q$ term in this expression does not appear in our final expression (32) for the fluctuations; rather, these are controlled entirely by the damping term $\Gamma(\mathbf{q}) + 2\alpha\left(\frac{q_y}{q}\right)^2$. Balancing $\omega$ against that term in the dominant regime of wavevector $q_x \sim q_x^2$ gives $\omega \propto q_x^2$, which implies $z = 2$.

Now we seek $\chi_y$, which determines whether or not the ordered state is stable against fluctuations in an arbitrarily large system. This can be obtained by looking at the real space fluctuations $\langle u_y^2(\mathbf{r},t) \rangle = \int_{q_x \gtrsim \frac{1}{L}} \frac{d^2 q}{(2\pi)^2} \langle |u_y(\mathbf{q},t)|^2 \rangle$, where $L$ is the lateral extent of the system in the $x$ direction (its extent in the $y$ direction is taken for the purposes of this argument to be infinite). Using (32), this integral is readily seen to converge in the infra-red, and, hence, as system size $L \to \infty$. Since the integral is finite, and proportional to the noise strength $D$, it is clear that, for sufficiently small $D$, the transverse $u_y$ fluctuations in real space can be made small enough that long-ranged orientational order, and, hence, a non-zero $\langle \mathbf{v}(\mathbf{r},t) \rangle$, is preserved in the presence of fluctuations; the ordered state is stable against fluctuations for sufficiently small mean noise strength $D$.

The exponent $\chi_y$ can be obtained by looking at the departure $\delta u_y^2$ of the $u_y$ fluctuations from their infinite system limit: $\delta u_y^2 \equiv \langle u_y^2(\mathbf{r},t) \rangle|_{L=\infty} - \langle u_y^2(\mathbf{r},t) \rangle|_L = \int_{q_x \lesssim \frac{1}{L}} \frac{d^2 q}{(2\pi)^2} \langle |u_y(\mathbf{q},t)|^2 \rangle$; we define the 'roughness exponent' $\chi_y$ by the way this quantity scales with system size $L$: $\delta u_y^2 \propto L^{2\chi_y}$ Note that this definition of $\chi_y$ requires $\chi_y < 0$, since it depends on the existence of an ordered state, which necessarily implies that the velocity fluctuations $\delta u_y^2$ do not diverge as $L \to \infty$. If $\langle u_y^2(\mathbf{r},t) \rangle|_{L=\infty}$ is not finite, one can obtain $\chi_y$ by performing exactly the type of scaling argument outlined here directly on $\langle u_y^2(\mathbf{r},t) \rangle|_L$ itself.

Approximating (32) for the dominant regime of wavevector $q_y \sim q_x^2$, and changing variables in the integral from $q_{x,y}$ to $Q_{x,y}$ according to $q_x \equiv \frac{Q_x}{L}$, $q_y \equiv \frac{Q_y}{L^2}$ shows that $\delta u_y^2 \propto L^{-1}$, and hence $\chi_y = -\frac{1}{2}$.

Note also that the fluctuations of $u_x$ are much smaller than those of $u_y$. This can be seen by using the incompressibility condition, which implies, in Fourier space, $u_x = -\frac{q_y u_y}{q_x}$, which implies

$$\left\langle |u_x(\mathbf{q},t)|^2 \right\rangle = \frac{Dq_y^2}{2\alpha q_y^2 + \Gamma(\mathbf{q})q^2} \approx \frac{Dq_y^2}{2\alpha q_y^2 + \mu q_x^4}, \quad (33)$$

which is clearly finite as $\mathbf{q} \to 0$ along any locus; indeed, it is bounded above by $\frac{D}{2\alpha}$.

We can calculate a roughness exponent $\chi_x$ for $u_x$ for the linear theory from this result exactly as we calculate the roughness exponent $\chi_y$ for $u_y$; we find $\chi_x = 1 - \zeta + \chi_y = -\frac{3}{2}$. We shall see in the next section that the first line of this equality also holds in the full non-linear theory, even though the values of the exponents $\chi_x$, $\zeta$, and $\chi_y$ all change.

The fact that $u_x$ has much smaller fluctuations than $u_y$ means that we have to work to higher order in $u_y$ than in $u_x$ when we treat the non-linear theory, as we do in next section.

**Mapping to an equilibrium 'divergence-free' magnet.** We now go beyond the linear theory, and expand the full equation of motion (6) of the main text to higher order in $\mathbf{u}$. We obtain

$$\partial_t u_m = -\partial_m P - 2\alpha u_x \delta_{mx} - \lambda_1^0 v_0 \partial_x u_m + \mu_T^0 \nabla^2 u_m + \mu_2^0 v_0^2 \partial_x^2 u_m + f_m \\ - \frac{\alpha}{v_0}\left[\frac{u_y^3}{v_0}\delta_{my} + 2u_x u_y \delta_{my} + u_y^2 \delta_{mx}\right] - \lambda_1^0 u_y \partial_y u_m \delta_{my}. \quad (34)$$

We keep terms that might naively appear to be higher order in the small fluctuations (for example, the $u_y^3 \delta_{my}$ term relative to the $u_x u_y \delta_{my}$ term) because, as we saw in the linearized theory, the two different components $u_{x,y}$ of $\mathbf{u}$ scale differently at long-length scales. Hence, it is not immediately obvious, for example, which of the two terms just mentioned is actually most important at long distances. We therefore, for now, keep them both. For essentially the same reason, it is not

obvious whether $u_y^2 \delta_{mx}$ or $u_y^3 \delta_{my}$ is more important, so we shall for now keep both of these terms as well.

On the other hand, it is immediately obvious that a term like, for example, $u_x u_y^2 \delta_{mx}$ is less relevant than $u_y^2 \delta_{mx}$, since, whatever the relative scaling of $u_x$ and $u_y$, $u_x u_y^2 \delta_{mx}$ is much smaller at large distances than $u_y^2 \delta_{mx}$, since $u_x$ is small.

Likewise, we drop the term $\frac{1}{2} \left(\frac{d\lambda_1}{dv}\right)_{v=v_0} u_y^2 \partial_x u_y \delta_{my}$, since it is manifestly smaller, by one $\partial_x$, than the $u_y^3 \delta_{my}$ term already displayed explicitly in (34).

This sort of reasoning guides us very quickly to the reduced model (34). As explained in the main text, acting on both sides of (34) with the transverse projection operator $P_{lm}(\mathbf{q}) = \delta_{lm} - q_l q_m / q^2$ which projects orthogonal to the spatial wavevector $\mathbf{q}$ eliminates the pressure term. Then taking the $l = y$ component of the resulting equation gives (11) of the main text, which we now use to calculate the rescaled coefficients.

To do this, we must also determine how the projection operators $P_{yx}$ and $P_{yy}$ rescale on the rescalings (that is, (12) of the main text). Since in the linear theory (see, for example, the $u_y - u_y$ correlation function (32)) fluctuations are dominated by the regime $q_y \lesssim q_x^2$, it follows that $P_{yx}(\mathbf{q}) = -\frac{q_x q_y}{q^2} \approx -q_y/q_x \ll 1$ and $P_{yy}(\mathbf{q}) = 1 - \frac{q_y^2}{q^2} \approx 1$. This implies that these rescale according to

$$P_{yx}(\mathbf{q}) \mapsto e^{(1-\zeta)\ell} P_{yx}(\mathbf{q}), \quad P_{yy}(\mathbf{q}) \mapsto P_{yy}(\mathbf{q}). \tag{35}$$

Performing the rescalings (12–14) of the main text, and (35) above on the equation of motion (11) of the main text, we obtain, from the rescalings of first three (that is, the linear) terms on the right-hand side the following rescalings of the parameters:

$$\nu_1 \mapsto e^{(z-1)\ell} \nu_1, \quad \mu \mapsto e^{(z-2)\ell} \mu, \\ \mu_T^0 \mapsto e^{(z-2\zeta)\ell} \mu_T^0, \tag{36}$$

and

$$\alpha \mapsto e^{(z-2\zeta+2)\ell} \alpha. \tag{37}$$

Note that the $\Gamma(\mathbf{q})$ term in (11) of the main text involves two parameters ($\mu$ and $\mu_T^0$); hence, we get the rescalings of both of these parameters from this term.

Similarly, looking at the rescaling of the non-linear terms proportional to $u_y^2$ and $u_y^3$, respectively, we obtain the rescalings:

$$\frac{\alpha}{\nu_0} \mapsto e^{(z+\chi_y-\zeta+1)\ell} \frac{\alpha}{\nu_0}, \quad \frac{\alpha}{\nu_0^2} \mapsto e^{(z+2\chi_y)\ell} \frac{\alpha}{\nu_0^2}. \tag{38}$$

We recover the first of these by looking at the rescaling of the non-linear term proportional to $u_x u_y$ as well.

We note that the two rescalings (38) are both consistent with (37) if we rescale $\nu_0$ according to

$$\nu_0 \mapsto e^{(1-\zeta-\chi_y)\ell} \nu_0. \tag{39}$$

By power counting on the $u_y \partial_y u_y$ term, we obtain the rescaling of $\lambda_1^0$:

$$\lambda_1^0 \mapsto e^{(z+\chi_y-\zeta)\ell} \lambda_1^0. \tag{40}$$

Finally, by looking at the rescaling of the noise correlations (that is, (4) of the main text), we obtain the scaling of the noise strength $D$:

$$D \mapsto e^{(z-2\chi_y-\zeta-1)\ell} D. \tag{41}$$

We now use the standard renormalization group logic to assess the importance of the non-linear terms in (11) of the main text. This logic is to choose the rescaling exponents $z$, $\zeta$ and $\chi_y$ so as to keep the size of the fluctuations in the field $\mathbf{u}$ fixed on rescaling. Since, as we saw in our treatment of the linearized theory (in particular, equation (32)), that size is controlled by three parameters: the 'longitudinal mass' $\alpha$, the damping coefficient $\mu$, and the noise strength $D$, the choice of $z$, $\zeta$ and $\chi_y$ that keeps these fixed will clearly accomplish this. From the rescalings (36), (37) and (41), this leads to three simple linear equations in the three unknown exponents $z$, $\zeta$ and $\chi_y$; solving these, we find the values of these exponents in the linearized theory:

$$\zeta_{\text{lin}} = z_{\text{lin}} = 2, \quad \chi_{y\text{lin}} = -1/2, \quad \chi_{x\text{lin}} = -3/2 \tag{42}$$

which, unsurprisingly, are the linearized exponents we found earlier.

With these exponents in hand, we can now assess the importance of the non-linear terms in (11) of the main text at long-length scales, simply by looking at how their coefficients rescale. (We do not have to worry about the size of the actual non-linear terms themselves changing on rescaling, because we have chosen the rescalings to keep them constant in the linear theory.) The mass $\alpha$, of course, is kept fixed. Inserting the linearized exponents (42) into the rescaling relation (39) for $\nu_0$, we see that

$$\nu_0 \mapsto e^{-\frac{\ell}{2}} \nu_0. \tag{43}$$

Since $\nu_0$ appears in the denominator of all three of the non-linear terms associated with $\alpha$, and $\alpha$ itself is fixed, this implies that all three of those terms are 'relevant', in the renormalization group sense of growing larger as we go to longer wavelengths (that is, as $\ell$ grows). As usual in the renormalization group, this implies that these terms ultimately alter the scaling behaviour of the system at sufficiently long distances. In particular, the exponents $z$, $\zeta$ and $\chi_{x,y}$ change from their values (42) predicted by the linear theory.

The same is not true of the $\lambda_1^0$ non-linearity, however, because it is irrelevant; that is, it gets smaller on renormalization. This follows from inserting the linearized exponents (42) into the rescaling relation (40) for $\lambda_1^0$, which gives

$$\lambda_1^0 \mapsto e^{-\frac{\ell}{2}} \lambda_1^0, \tag{44}$$

which shows clearly that $\lambda_1^0$ vanishes as $\ell \to \infty$; that is, in the long-wavelength limit.

Since $\lambda_1^0$ was the only remaining non-linearity associated with the $\lambda$ terms in our original equation of motion (34), we can accurately treat the full, long-distance behaviour of this problem by leaving out all of those non-linear terms. Doing so reduces the equation of motion (34) to

$$\partial_t u_m = -\lambda_1^0 \nu_0 \partial_x u_m - \partial_m P - 2\alpha \left(u_x + \frac{u_y^2}{2\nu_0}\right) \delta_{xm} \\ - \frac{2\alpha}{\nu_0} \left(u_x + \frac{u_y^2}{2\nu_0}\right) u_y \delta_{ym} \\ + \mu \partial_x^2 u_m + \mu_T^0 \partial_y^2 u_m + f_m. \tag{45}$$

Before proceeding to analyse this equation, we note the differences between the structure of this problem and that of the compressible case. In the compressible problem, there is no constraint analogous to the incompressibility condition relating $u_x$ and $u_y$. Hence, $u_x$ is free to relax quickly (to be precise, on a time scale $\frac{1}{2\alpha}$) to its local 'optimal' value, which is readily seen to be $-\frac{u_y^2}{2\nu_0}$. Once this relaxation has occurred, all of the non-linearities associated with $\alpha$ drop out of that compressible problem, leaving the $\lambda$ non-linearities as the dominant ones. For a detailed discussion of the rather tricky analysis of the compressible problem that leads to this conclusion, see ref. 41. Here, in the incompressible problem, $u_x$ is, because of the incompressibility constraint, not free to relax in such a way as to cancel out the $\alpha$ non-linearities, which, because they involve no spatial derivatives, wind up dominating the $\lambda$ non-linearities, which do involve spatial derivatives. In addition, the suppression of fluctuations by the incompressibility condition, which as we have already seen in the linear theory, makes the $\lambda$ non-linearities not only less relevant than the $\alpha$ ones, but actually irrelevant. Hence, we can drop them in this incompressible problem, leaving us with equation (45) as our equation of motion.

As one final simplification, we make a Galilean transformation to a 'pseudo-co-moving' co-ordinate system moving in the direction $\hat{x}$ of mean flock motion at speed $\lambda_1^0 \nu_0$. Note that if the parameter $\lambda_1^0$ had been equal to 1, this would be precisely the frame co-moving with the flock. The fact that it is not is a consequence of the lack of Galilean invariance in our problem.

This boost eliminates the 'convective' term $\lambda_1^0 \nu_0 \partial_x u_m$ from the right-hand side of (45), leaving us with our final simplified form for the equation of motion:

$$\partial_t u_m = -\partial_m P - 2\alpha \left(u_x + \frac{u_y^2}{2\nu_0}\right) \delta_{xm} \\ - \frac{2\alpha}{\nu_0} \left(u_x + \frac{u_y^2}{2\nu_0}\right) u_y \delta_{ym} \\ + \mu \partial_x^2 u_m + \mu_T^0 \partial_y^2 u_m + f_m. \tag{46}$$

which is just equation (15) of the main text.

**Mapping of equilibrium 'divergence-free' magnet to 2D smectic.** We begin by demonstrating the pictorial interpretation of the 'streaming function' introduced in the main text via

$$u_x = -\nu_0 \partial_y h, \quad u_y = \nu_0 \partial_x h. \tag{47}$$

This implies that the streaming function $\phi$ for the full velocity field $\mathbf{v}(\mathbf{r}) = \nu_0 \hat{x} + \mathbf{u}$, defined via $v_x = \partial_y \phi$, $v_y = -\partial_x \phi$, is given by

$$\phi = \nu_0 (y - h(\mathbf{r})). \tag{48}$$

As in conventional 2d fluid mechanics, contours of the streaming function $\phi$ are flow lines. When the system is in its uniform steady state (that is, $\mathbf{v} = \nu_0 \hat{x}$), these contour lines, defined via

$$\phi = nC, \quad n = 0, 1, 2, 3\ldots \tag{49}$$

where $C$ is some arbitrary constant, are a set of parallel, uniformly spaced lines given by $y_n = nC/\nu_0$.

Now let us ask what the flow lines are if there are fluctuations in the velocity field: $\mathbf{v} = \nu_0 \hat{x} + \mathbf{u}$. Combining our expression for $\phi$ (48) and the expression (49) for the flow lines, we see that the positions of the flow lines are now given by

$$y_n = nC/\nu_0 + h, \quad n = 0, 1, 2, 3\ldots, \tag{50}$$

which shows that $h(\mathbf{r})$ can be interpreted as the local displacement of the flow lines from their positions in the ground state configuration.

This picture of a set of lines that 'wants' to be parallel being displaced by a fluctuation $h(\mathbf{r})$ looks very much like a 2D smectic liquid crystal (that is, 'soap'), for which the layers are actually one-dimensional fluid stripes.

This resemblance between our system and a 2D smectic is not purely visual. Indeed, making the substitution (47), the Hamiltonian (17) of the main text becomes (ignoring irrelevant terms like $(\partial_x \partial_y h)^2$, which is irrelevant compared with $(\partial_x^2 h)^2$ because $y$-derivatives are less relevant than $x$-derivatives)

$$H_s = \frac{1}{2} \int d^2 r \left[ B \left( \partial_y h - \frac{(\partial_x h)^2}{2} \right)^2 + K(\partial_x^2 h)^2 \right],$$  (51)

where $B = 2\alpha v_0^2$ and $K = \mu v_0^2$. This Hamiltonian is exactly the Hamiltonian for the equilibrium 2D smectic model with $h$ in equation (51) interpreted as the displacement field of the smectic layers. For the equilibrium 2D smectic the partition function is

$$Z_s = \int D[h] e^{-H_s/k_B T},$$  (52)

where it should be noted that there is no constraint on the functional integral over $h(\mathbf{r})$ in this expression, since, as noted earlier, $h(\mathbf{r})$ is unconstrained.

Since the variable transformation equation (47) is linear, the partition functions for the smectic: $Z_s$ (equation (52)) and that for the constrained $XY$ model:

$$Z_{XY} = \int D[u_x] D[u_y] \delta(\nabla \cdot \mathbf{u} = 0) e^{-H_{XY}/k_B T},$$  (53)

are the same up to a constant Jacobian factor, which changes none of the statistics.

To summarize what we have learned so far, we have successfully mapped the model for the ordered phase of an incompressible polar active fluid onto the ordered phase of the equilibrium 2D $XY$ model with the constraint $\nabla \cdot \mathbf{u} = 0$, which in turn we have mapped onto the standard equilibrium 2D smectic model[30]. The scaling behaviours of the former can therefore be obtained by studying the latter. Note that the connection between our problem and the dipolar magnet, which was studied in ref. 30, is that the long-ranged dipolar interaction in magnetic systems couples to, and therefore suppresses, the longitudinal component of the magnetization. See ref. 30 for more details.

**Mapping the 2D smectic to the $(1+1)$-D KPZ equation.** Fortunately, the scaling behaviours of the equilibrium 2D smectic model are known, thanks to an ingenious further mapping[21,22] of this problem onto the $1+1$-dimensional KPZ equation[8], which is a model for interface growth or erosion (for example, 'sandblasting'). In this mapping, which connects the equal-time correlation functions of the 2D smectic to the $1+1$-dimensional KPZ equation, the $y$-coordinate in the smectic is mapped onto time $t$ in the $1+1$-dimensional KPZ equation with $h(x, t)$ the height of the 'surface' at position $x$ and time $t$ above some reference height. As a result, the dynamical exponent $z_{KPZ}$ of the $1+1$-dimensional KPZ equation becomes the anisotropy exponent $\zeta$ of the 2d smectic. Since the scaling laws of the $1+1$-dimensional KPZ equation are known exactly[8], those of the equal-time correlations of the 2D smectic can be obtained as well.

This gives[21,22] $\zeta = 3/2$ and $\chi_h = 1/2$ as the exponents for the 2D smectic, where $\chi_h$ gives the scaling of the smectic layer displacement field $h$ with spatial coordinate $x$. Given the streaming function relation (47) between $h$ and $\mathbf{u}$, we see that the scaling exponent $\chi_y$ for $u_y$ is just $\chi_y = \chi_h - 1 = -1/2$ and that the scaling exponent $\chi_x$ for $u_x$ is just $\chi_x = \chi_y + 1 - \zeta = -1$.

The fact that both of the scaling exponents $\chi_y$ and $\chi_x$ are less than zero implies that both $u_y$ and $u_x$ fluctuations remain finite as system size $L \to \infty$; this, in turn, implies that the system has long-ranged orientational order. That is, the ordered state is stable against fluctuations, at least for sufficiently small noise $D$.

The velocity correlation function can be calculated through the connection between $\mathbf{u}$ and $h$. Using the aforementioned connection between 2D smectics and the $1+1$-dimensional KPZ equation, the layer displacement correlation function takes the form[21,22]:

$$C_h(\mathbf{r} - \mathbf{r}') \equiv \left\langle [h(\mathbf{r}, t) - h(\mathbf{r}', t)]^2 \right\rangle$$
$$= B|x - x'| \Psi(\kappa).$$  (54)

where the scaling variable $\kappa \equiv \frac{X}{Y^{2/3}}$, $X = |x - x'|/\xi_x$, and $Y = |y - y'|/\xi_y$, $B$ is a non-universal overall multiplicative factor extracted from the scaling function, and the non-universal nonlinear lengths $\xi_{x,y}$ are calculated in the next section. The universal scaling function $\Psi$ has been numerically estimated (www-m5.ma.tum.de/KPZ)[37–40]:

$$\Psi(\kappa) \approx \begin{cases} c_1 + e^{-\Phi_h(\kappa)}, & \kappa \gg 1 \\ \kappa + \frac{c_2}{\kappa}, & \kappa \ll 1, \end{cases}$$  (55)

where for $\kappa \gg 1$,

$$\Phi_h(\kappa) = c\kappa^3 + \mathcal{O}(\kappa).$$  (56)

Here, the constants $c$ and $c_{1,2}$ are all universal and are given by $c \approx 0.295$, $c_1 \approx 1.843465$ and $c_2 \approx 1.060\ldots$ (www-m5.ma.tum.de/KPZ)[39,40]. Only the lengths $\xi_{x,y}$ and the overall multiplicative factor of $B$ in (54) are non-universal (that is, system-dependent).

Rewriting the velocity correlation function (equation (1) of the main text) in terms of the fluctuations $\mathbf{u}$ of the velocity from its mean value (as defined in equation (7) of the main text), we find

$$\left\langle |\mathbf{v}(\mathbf{r}, t) - \mathbf{v}(\mathbf{r}', t)|^2 \right\rangle$$
$$= C_0 - 2\langle u_y(\mathbf{r}, t) u_y(\mathbf{r}', t) \rangle - 2\langle u_x(\mathbf{r}, t) u_x(\mathbf{r}', t) \rangle,$$  (57)

where $C_0 = 2\langle |\mathbf{u}(\mathbf{r}, t)|^2 \rangle$ is finite. The two correlation functions on the right-hand side of the equality are just the derivatives of the layer displacement correlation function:

$$\langle u_y(\mathbf{r}, t) u_y(\mathbf{r}', t) \rangle = -\frac{v_0^2}{2} \partial_x \partial_{x'} C_h(\mathbf{r} - \mathbf{r}')$$
$$= \frac{Bv_0^2}{2|x - x'|} \Psi_y(\kappa),$$  (58)

$$\langle u_x(\mathbf{r}, t) u_x(\mathbf{r}', t) \rangle = -\frac{v_0^2}{2} \partial_y \partial_{y'} C_h(\mathbf{r} - \mathbf{r}')$$
$$= \frac{Bv_0^2 \xi_x^3}{9\xi_y^2 |x - x'|^2} \Psi_x(\kappa).$$  (59)

The velocity component scaling functions $\Psi_{x,y}$ can both be expressed in terms of the height scaling function $\Psi$, via

$$\Psi_y(\kappa) = \kappa(2\Psi' + \kappa\Psi''),$$  (60)

$$\Psi_x(\kappa) = \kappa^4(5\Psi' + 2\kappa\Psi'').$$  (61)

Using the asymptotic forms (55) for the height scaling function $\Psi$ in (60) and (61), we obtain the asymptotic behaviours:

$$\Psi_y(\kappa) \approx \begin{vmatrix} 2\kappa, & \kappa \ll 1 \\ 9c^2 \kappa^6 e^{-\Phi_h(\kappa)}, & \kappa \gg 1, \end{vmatrix}$$  (62)

$$\Psi_x(\kappa) \approx \begin{cases} -c_3 \kappa^2, & \kappa \ll 1, \\ 18c^2 \kappa^9 e^{-\Phi_h(\kappa)}, & \kappa \gg 1. \end{cases}$$  (63)

Using these expressions (62, 63) for the scaling functions in the scaling expressions (58, 59) for the $u_x$ and $u_y$ correlation functions, and in turn using those in our expression (57) for the velocity correlation function, we obtain

$$\left\langle |\mathbf{v}(\mathbf{r}, t) - \mathbf{v}(\mathbf{r}', t)|^2 \right\rangle$$
$$= \begin{cases} C_0 - AY^{-2/3}, & \kappa \ll 1 \\ C_0 - \frac{9}{2} c^2 \frac{A}{X} e^{-\Phi(\kappa)} \left[ 1 + \frac{4}{9} \left( \frac{x - x'}{y - y'} \right)^2 \right], & \kappa \gg 1 \end{cases}$$  (64)

where the non-universal constant $A$ is given by

$$A = 2Bv_0^2 \xi_x^{-1}.$$  (65)

We have also defined a new universal function

$$\Phi(\kappa) \equiv \Phi_h(\kappa) - 6\ln(\kappa),$$  (66)

which has the same limiting behaviour as $\Phi_h$, namely,

$$\Phi(\kappa) = c\kappa^3 + \mathcal{O}(\kappa).$$  (67)

since the additive logarithm in (67) is sub-dominant to the leading $\kappa^3$ term.

**Calculation of the nonlinear lengths.** The nonlinear lengths $\xi_{x,y}$ can be calculated most conveniently from the equilibrium 2D smectic model (51). By definition, $\xi_x$ and $\xi_y$ are the lengths along $x$ and $y$ beyond which the anharmonic terms in equation (51) become important. To determine these lengths, we treat the anharmonic terms perturbatively, and calculate the lowest order correction to the harmonic terms. In a finite system of linear dimensions $L_{x,y}$, this 'perturbative' correction will indeed be perturbative (that is, small) compared with the 'bare' values of the harmonic terms. However, they grow without bound with increasing $L_{x,y}$, and, hence, eventually cease to be small; that is, the perturbation theory breaks down at large $L_{x,y}$. The values of $L_{x,y}$ above which the perturbation theory breaks down are the nonlinear lengths $\xi_{x,y}$.

Calculating the lowest order correction to the compression modulus $B$ (that is, the coefficient of $(\partial_y h)^2$ in the smectic Hamiltonian (51)) can be graphically represented by the Feynman diagram in Fig. 3. This leads to a correction to compression modulus:

$$\delta B = -\frac{k_B T B^2}{(2\pi)^2} \int_{\frac{1}{L_x}}^\infty dq_x \int_{-\infty}^\infty dq_y \frac{q_x^4}{\left(Bq_y^2 + Kq_x^4\right)^2}$$
$$= -\frac{k_B T}{8\pi} \left(\frac{B}{K}\right)^{\frac{3}{2}} \int_{\frac{1}{L_x}}^\infty dq_x \frac{1}{q_x^2}$$
$$= -\frac{k_B T}{8\pi} \left(\frac{B}{K}\right)^{\frac{3}{2}} L_x.$$  (68)

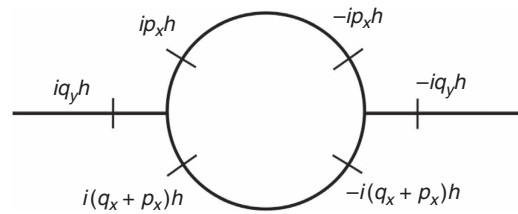

**Figure 3 | Feynman graph for the compression modulus (B).** This arises from the combination of two cubic terms.

In this calculation, we have taken $L_y$, the system size along $y$, to be infinite. By the definition of $\xi_x$, $|\delta B| = B$ for $L_x = \xi_x$, which gives

$$\xi_x = \frac{8\pi B}{k_B T}\left(\frac{K}{B}\right)^{\frac{3}{2}} = \frac{4\sqrt{2}\pi v_0^2 \mu^{\frac{3}{2}}}{D\alpha^{\frac{1}{2}}}, \tag{69}$$

where in the second equality we have used the relations $B = 2\alpha v_0^2$, $K = \mu v_0^2$, and $k_B T = D$ between the parameters of the smectic and those of the original incompressible active fluid.

Likewise, doing the same calculation for $L_y = \xi_y$, $L_x = \infty$, we find

$$\xi_y = \frac{64\sqrt{2}\pi^2 v_0^4 \mu^{\frac{5}{2}}}{D^2 \alpha^{\frac{1}{2}}}. \tag{70}$$

**Fourier transformed correlation functions.** The spatially Fourier transformed autocorrelations are also of interest. Fourier transforming (54) gives

$$\langle |h(\mathbf{q},t)|^2 \rangle = -\frac{1}{2}\int dx\,dy\, e^{i(q_x x + q_y y)} |x| \Psi\left(\frac{\left(\frac{|x|}{\xi_x}\right)}{\left(\frac{|y|}{\xi_y}\right)^{\frac{2}{3}}}\right). \tag{71}$$

With the change of variables to new variables of integration $S$ and $W$ via $x \equiv \frac{S}{q_x}$ and $y \equiv \frac{W\xi_y}{(q_x\xi_x)^{3/2}}$, we immediately obtain

$$\langle |h(\mathbf{q},t)|^2 \rangle = q_x^{-7/2} f\left(q_y/q_x^{3/2}\right), \tag{72}$$

with

$$f(\Theta) \equiv -\frac{\xi_y}{2\xi_x^{3/2}}\int dS\,dW\, e^{i(S+\Theta W)} |S| \Psi\left(\frac{|S|}{|W|^{\frac{2}{3}}}\right). \tag{73}$$

Combining equation (72) with the Fourier transform of the variable transformation (19) of the main text, we obtain the correlation functions for the ordered phase of the constrained equilibrium 2D $XY$ model, and hence, the ordered phase of incompressible active fluids:

$$\langle |u_y(\mathbf{q},t)|^2 \rangle = q_x^2 \langle |h(\mathbf{q},t)|^2 \rangle = q_x^{-\frac{3}{2}} f\left(q_y/q_x^{3/2}\right) \tag{74}$$

$$\begin{aligned}\langle |u_x(\mathbf{q},t)|^2 \rangle &= q_y^2 \langle |h(\mathbf{q},t)|^2 \rangle = q_y^2 q_x^{-\frac{7}{2}} f\left(q_y/q_x^{3/2}\right) \\ &= q_x^{-\frac{1}{2}} g\left(q_y/q_x^{3/2}\right),\end{aligned} \tag{75}$$

where $g(w) \equiv w^2 f(w)$. The limiting behaviours of the scaling functions $f(x)$ and $g(x)$ are: $f(x\to 0) \to \text{constant} \neq 0$, $f(x\to\infty) \propto x^{-7/3}$, $g(x\to 0) \propto x^2$ and $g(x\to\infty) \propto x^{-1/3}$.

**Data availability.** The data that support the findings of this study are available from any of the corresponding authors on request.

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

## Acknowledgements

We thank Pawel Romancsuk for pointing out the geometrical interpretation of the field $h(\mathbf{r})$ as the displacement of the flow lines, and Herbert Spohn for referring

us to ref.40. J.T. also thanks the Max Planck Institute for the Physics of Complex Systems in Dresden, Germany, the Department of Bioengineering at Imperial College, London, the Kavli Institute for Theoretical Physics, Santa Barbara, CA, and the Lorentz Center of Leiden University, for their hospitality while this work was underway. He also thanks the US NSF for support by awards # EF-1137815 and 1006171; and the Simons Foundation for support by award #225579. L.C. acknowledges support by the National Science Foundation of China (under Grant No. 11474354).

## Author contributions

All authors contributed equally to this work.

## Additional information

**Competing financial interests:** The authors declare no competing financial interests.

