## [Peer Review File · Nature Communications]

Reviewers' comments:

Reviewer #1 (Remarks to the Author):

The authors demonstrate the presence of long-range ordering in 2d active fluids by a novel sequence of mappings of the hydrodynamic theory of incompressible active fluids onto i) the divergence-free XY model, which can be casted into ii) a 2d smectic liquid crystal formulation by introducing the streaming function. In turn, using previous work, the smectic liquid crystal model is connected to the $(1+1)$ -dimensional KPZ equation for growing interfaces. The KPZ interfaces are known to exhibit long-range order, thus certain robustness against finite fluctuations. Using this mapping to KPZ, the authors are able to extract the scaling exponents of the equal-time velocity correlation function and compare them with those determined by the linearized equations. The overall presentation is clear and the analysis seems correct. There is a fair amount of references, the background work could be better emphasized.

This is an interesting study which map spawn future theoretical and experimental work to further study the universality class connecting these different systems. I recommend publications after minor revisions regarding few comments.

The hydrodynamic theory and its connection to the XY model has already been studied in Ref. [11,12]. It seems to me that the incompressible limit is crucial for the connection to the KPZ equation particularly that the connection to the XY model was already known before. Therefore, it would be useful to discuss more clearly the importance of the incompressible limit. For instance, the long-range order is also present in the compressible active fluids, so how is that different from the incompressible case? Are the scaling exponents different? How is the mapping onto KPZ affected by the incompressibility condition?

The background work could be discussed more clearly in the introduction/conclusion, in particular which connections have already been made previously and what this paper is adding to that.

Reviewer #2 (Remarks to the Author):

In their paper, Chen et al. establish an interesting, previously unknown mathematical correspondence that connects an incompressible active fluid model with members of the KPZ universality class. Nontrivial analytical results in active matter theory are rare and I therefore believe that the important scaling laws derived in the paper deserve to be published in Nature Communications. I have, however, several comments and questions that that I believe should be addressed by the authors prior to acceptance of the article.

1) The authors start their introduction by emphasizing the differences between active and equilibrium systems. Yet, their derivation of the critical exponents relies crucially on the fact that they can map their non-equilibrium system onto an incompressible 2D XY model (or LC model) at equilibrium.

I find parts of the overall representation a bit confusing in this regard. In my opinion, it would be helpful for the broad readership if the authors could explain/summarize more clearly in the introduction/conclusions which parts of their results exactly manifest the universal analogy and the fundamental differences between the non-equilibrium flocking/KPZ systems and the corresponding equilibrium XY/LC systems.

2) The authors emphasize in the introduction that they focus on incompressible systems and support this restriction by providing a number convincing experimental examples (see also PRL 110: 228102, 2013). It seems therefore redundant to present first the compressible model in Eq. (2) and (3), which does not really have any relevance for the results of the paper. Starting the Model section with (5) and (6) seems completely sufficient and would also remove the unnecessary reference to unpublished future work.

3) The authors consider the simplest possible form of noise, corresponding to an additive Gaussian random force - which I think is fair given the purpose of their study. However, the coupling term for such external noise differs substantially from that of intrinsic stress-based noise in conventional fluctuating hydrodynamics (see e.g. Landau-Lifshits).

Is it correct to assume that the scaling relations/exponents may change significantly in the latter case? Can the authors' arguments be generalized to additive noise in the stress-tensor? If not, where does the argument fail?

I would recommend that the authors add a few additional explanatory remarks that clarify the physical assumptions/differences underlying their noise model relative to noise in fluctuating HD models.

4) Another question that deserves to be discussed in more detail, in my opinion, concerns the general validity of Eq. (6) in the incompressible high-density regime. Recent experimental studies of bacteria and simulations of polar active matter report equal-time velocity correlations functions (e.g., PNAS 109: 14308, 2012; PRL110: 228102, 2013) that are negative on intermediate length scales, reflecting the emergence/ presence of vortices in these systems. These findings suggest that, at least for certain classes of dense active systems, higher-order derivative terms might have to be added in Eq. (6).

I believe it would be beneficial for the reader, if the authors added a few sentences that state more clearly potential limits on the applicability of Eq. (6).

Other minor comments/suggestions:

1. page 2, col 2, line 3-5:

Is there a qualifier ("algebraically" or similar) missing in this sentence?

If not, why would an arbitrarily decaying correlation function indicate long-range order?

2. In the broader context of their analysis, the authors may find this recent experimental study of interest

<http://www.nature.com/nphys/journal/vaop/ncurrent/full/nphys3607.html>

3. typos in conclusions section:

relevant informations

mathematical transformations

4. Methods section, before equation (18):

What is "ouch!" supposed to mean?

Reviewer #3 (Remarks to the Author):

This is a very interesting paper on connections amongst several seemingly distinct models of ordering and the breaking of continuous rotational symmetry in stationary states far from thermal equilibrium. The authors show that the equal-time statistical properties of the ordered phase of an incompressible flock in two dimensions are related to the height fluctuation statistics of the KPZ equation in $1 + 1$ dimension, which are known to be related to 2D smectics, and are shown further to be connected to a divergence-free XY model.

The paper is conceptually interesting, and a real understanding of the work places considerable technical demand on the reader. But this is because it is a very nontrivial piece of physics. Moreover, this is not arid theory: it makes sharp predictions for experiment. The main results and the nice pictorial interpretation of some parts of the calculation should appeal to the very broad readership that Nat Comm would like to address. I recommend acceptance once my questions are answered satisfactorily.

The authors consider hydrodynamic flocking models in which the bulk modulus for the isotropic part of the pressure in the equation for the velocity vector field (the order parameter for this problem) goes to infinity, so that the velocity is solenoidal. A curious technical consequence is that the analysis even in the ordered phase requires keeping track of the components of the velocity along as well as transverse to the ordering direction participate significantly in the dynamics of the ordered phase. The dominant nonlinearity, moreover, turns out to be that governing the shape of the potential that controls the order-parameter magnitude, not the "advective" term. Both these results are in marked contrast to the situation prevailing in compressible flocking models. This is genuinely new physics introduced by the long-ranged constraint of incompressibility. The authors work out the consequences of these nonlinearities, ultimately via the connection to $1+1$ -dimensional KPZ.

I have two remarks to which I would like the authors to respond.

1) The ordered phase of colloidal rollers studied by Bricard et al., ref [26] of the paper under review, would appear to be a realization of an incompressible 2D flock. If so, how does the flock of the present work escape the bend instability of the Bricard work (which they mention only in Supplementary Information, after eq. S42)?

2) The authors say that the large-distance decay to zero of the connected part of the equal-time velocity correlator shows that long-range order prevails. As far as I can tell, it only shows that the full correlator $\langle v(r) \cdot v(r') \rangle$ approaches the disconnected part $\langle v \rangle \cdot \langle v \rangle$. It doesn't establish that the latter is nonzero. Am I missing something?

REVIEWERS' COMMENTS:

Reviewer #1 (Remarks to the Author):

The authors have addressed all my comments appropriately, and accordingly have improved the manuscript. I therefore recommend for publication.

Reviewer #2 (Remarks to the Author):

In their revised version and response letter, the authors have satisfactorily addressed my questions and comments. The only minor point I remain unconvinced of is that Eqs. (2) and (3) are indeed necessary. It seems to me the main purpose of these two equations is to advertise future work not included in the present manuscript. I believe that it would be completely sufficient for the present paper to start from Eqs. (5), (6) and (4) and a short statement that the authors focus on the non-jammed constant density limit, in which all coefficients depend on the constant density and pressure is reduced to a Lagrange-multiplier for the incompressibility constraint (5). But this is a minor point and I am happy to recommend the revised version for publication in Nature Communications.

Reviewer #3 (Remarks to the Author):

I am satisfied with the authors' response to my two remarks. I have also gone over their response to the points raised by the other referees and find their answers to be adequate. A small quibble: ref. [17] shows theoretically that spontaneous uniaxial order in (apolar, the title notwithstanding) active systems is *unstable* when the hydrodynamic interaction is included. I don't think a reference to this paper belongs where it's cited, in the context of realizations of polar active systems that could show long-range order. Apart from this the paper can be published in its present form.

Reviewer 1

The authors demonstrate the presence of long-range ordering in 2d active fluids by a novel sequence of mappings of the hydrodynamic theory of incompressible active fluids onto i) the diverge-free XY model, which can be casted into ii) a 2d smectic liquid crystal formulation by introducing the streaming function. In turn, using previous work, the smectic liquid crystal model is connected to the (1+1)-dimensional KPZ equation for growing interfaces. The KPZ interfaces are known to exhibit long-range order, thus certain robustness against finite fluctuations. Using this mapping to KPZ, the authors are able to extract the scaling exponents of the equal-time velocity correlation function and compare them with those determined by the linearized equations. The overall presentation is clear and the analysis seems correct. There is a fair amount of references, the background work could be better emphasized.

This is an interesting study which map spawn future theoretical and experimental work to further study the universality class connecting these different systems. I recommend publications after minor revisions regarding few comments.

Response: We are glad that the referee found our paper interesting and recommended publication. His/her comments will be addressed below.

The hydrodynamic theory and its connection to the XY model has already been studied in Ref. [11,12]. It seems to me that the incompressible limit is crucial for the connection to the KPZ equation particularly that the connection to the XY model was already known before. Therefore, it would be useful to discuss more clearly the importance of the incompressible limit. For instance, the long-range order is also present in the compressible active fluids, so how is that different from the incompressible case? Are the scaling exponents different? How is the mapping onto KPZ affected by the incompressibility condition?

Response: We agree with the referee that the incompressible limit is crucial for connecting our problem to the KPZ equation. Only in that limit can the hydrodynamic theory of active fluids be mapped onto $1 + 1$ KPZ equation. The referee is also correct in noting that long-ranged order is also present for compressible two dimensional flocks. The scaling exponents are, however, very different.

We have explained this more explicitly in the introduction, in the paragraph before equation (1), and immediately after we derive the exponents (specifically, two paragraphs before equation (21)).

The background work could be discussed more clearly in the introduction/conclusion, in particular which connections have already been made previously and what this paper is adding to that.

Response: We have now clearly stated the contribution of our paper in both the introduction and conclusion. Specifically, the connections between the ordered phase of 2D incompressible polar active fluids, the ordered phase of the divergence-free 2D XY model, and dislocation-free 2D smectics are all new and original to our paper.

Reviewer 2

In their paper, Chen et al. establish an interesting, previously unknown mathematical correspondence that connects an incompressible active fluid model with members of the KPZ universality class. Nontrivial analytical results in active matter theory are rare and I therefore believe that the important scaling laws derived in the paper deserve to be published in Nature Communications.

Response: We are glad that referee 2 believe our work deserves to be published in Nature Communications.

I have, however, several comments and questions that that I believe should be addressed by the authors prior to acceptance of the article.

1) The authors start their introduction by emphasizing the differences between active and equilibrium systems. Yet, their derivation of the critical exponents relies crucially on the fact that they can map their non-equilibrium system onto an incompressible 2D XY model (or LC model) at equilibrium.

I find parts of the overall representation a bit confusing in this regard. In my opinion, it would be helpful for the broad readership if the authors could explain/summarize more clearly in the introduction/conclusions which parts of their results exactly manifest the universal analogy and the fundamental differences between the non-equilibrium flocking/KPZ systems and the corresponding equilibrium XY/LC systems.

Response: We thank the referee for pointing out the potentially confusing issue. We have now modified the introduction and conclusion sections to clarify this issue. In particular, we have explicitly stated in the first paragraph that some (though not all) non-equilibrium systems belong to the same universality class as some equilibrium systems. Although the narrative of the manuscript is slightly changed as a result, all of our original conclusions remain valid.

Please see also our third response to Referee 1.

2) The authors emphasize in the introduction that they focus on incompressible systems and support this restriction by providing a number convincing experimental examples (see also PRL 110: 228102, 2013). It seems therefore redundant to present first the compressible model in Eq. (2) and (3), which does not really have any relevance for the results of the paper. Starting the Model section with (5) and (6) seems completely sufficient and would also remove the unnecessary reference to unpublished future work.

Response: There are *two* different cases for the incompressibility limit. In one case the number density of active particles is constant while the particles are still able to move. In the other case, the particles are jammed. To obtain the corresponding hydrodynamic model (i.e., Eqs. (5, 6)) for the former case, which we focus on in this paper, we have to start from the most general one (i.e., Eqs. (2, 3)), which applies to *both* compressible and incompressible polar active fluids, and take the appropriate incompressible limit as described by the text between Eqs. (2,3) and Eqs. (5, 6).

3) The authors consider the simplest possible form of noise, corresponding to an additive Gaussian random force - which I think is fair given the purpose of their study. However, the coupling term for such external noise differs substantially from that of intrinsic stress-based noise in conventional fluctuating hydrodynamics (see e.g. Landau-Lifshits).

Is it correct to assume that the scaling relations/exponents may change significantly in the latter case? Can the authors' arguments be generalized to additive noise in the stress-tensor? If not, where does the argument fail?

I would recommend that the authors add a few additional explanatory remarks that clarify the physical assumptions/differences underlying their noise model relative to noise in fluctuating HD models.

Response: We thank the referee for pointing out to us that this point needs to be clarified. While a “stress-based noise” of the type the referee proposes is definitely present in our problem, it is subdominant to the random force that we considered, since such a stress based forcing vanishes at zero wavevector (in Fourier space), while our random forcing does not. Such a zero-wavevector random forcing is not allowed in a momentum conserving system like an equilibrium fluid in free space (which is described by the Navier-Stokes equations), which is why it is not discussed in standard treatments in sources like Landau and Lifshitz. However, it *is* allowed in our system, in which momentum is NOT conserved, due to the presence of a substrate over which the flock moves. And once allowed, it dominates.

Note that such a term is also present in Foster, Nelson, and Stephens’ “model B”, which they also call the “randomly stirred fluid”, in which they consider random stirring that violates momentum conservation. Since we have no momentum conservation in our system, such a term is likewise allowed.

We have inserted a sentence below Eq. (4) to make these points clearer after our equation of motion, at the point at which we first discuss the random force.

4) Another question that deserves to be discussed in more detail, in my opinion, concerns the general validity of Eq. (6) in the incompressible high-density regime. Recent experimental studies of bacteria and simulations of polar active matter report equal-time velocity correlations functions (e.g., PNAS 109: 14308, 2012; PRL110: 228102, 2013) that are negative on intermediate length scales, reflecting the emergence/ presence of vortices in these systems. These findings suggest that, at least for certain classes of dense active systems, higher-order derivative terms might have to be added in Eq. (6). I believe it would be beneficial for the reader, if the authors added a few sentences that state more clearly potential limits on the applicability of Eq. (6).

Response: We agree with the referee that our model does not describe all possible types of collective motion. Indeed, we are limited to “ordered flocks”; that is, motion with long-ranged ferromagnetic correlations of the velocity (or, to put this another way, to situations in which the flock develops a non-zero average velocity). Such a state is only possible if the leading order damping terms (i.e., the diffusion coefficients $\mu_{T,B,2}$ in the equation of motion Eq. (3)) are positive. It was only because these coefficients proved to be negative in the bacterial experiments of PNAS 109: 14308, 2012; PRL110: 228102, 2013 (which did *not* see an ordered state) that those authors were forced to include higher order damping terms to stabilize the system. If those coefficients are positive, as they must be in an ordered system, then those higher order terms are not only unnecessary for stability, but are actually *irrelevant* to the long-distance properties of the system.

We have added a passage at the end of the first paragraph in the section Model to clarify this point.

Other minor comments/suggestions:

1. page 2, col 2, line 3-5: Is there a qualifier (“algebraically” or similar) missing in this sentence? If not, why would an arbitrarily decaying correlation function indicate long-range order?

Response: A qualifier is not necessary, and would, indeed, be incorrect. Provided that the correlation function as defined in our paper decays at large distance, the system has long-range orientational order. In fact, the decaying behavior of the correlation function in our paper is anisotropic, and does *not* decay algebraically in all directions. Although it *does* decay algebraically along y , it decays exponentially along x .

Since both this referee and referee 3 find the connection between the behavior of the original correlation function under discussion and long-range order (LRO) obscure, we have decided to replace it with the prediction for the more commonly used correlation function $\langle |\mathbf{v}(\mathbf{r}) - \mathbf{v}(\mathbf{r}')|^2 \rangle$ in Eq. (1). That this correlation function goes to finite value in the limit $|\mathbf{r} - \mathbf{r}'| \rightarrow \infty$ clearly implies LRO. The simplest argument for LRO, which is based on the negativity of the roughness exponents $\chi_{x,y}$, is now provided in the paragraph before that containing equation (21).

2. In the broader context of their analysis, the authors may find this recent experimental study of interest <http://www.nature.com/nphys/journal/vaop/ncurrent/full/nphys3607.html>

Response: We thank the referee for pointing us to this relevant paper. We now cite it in the introduction.

3. typos in conclusions section: relevant informations mathematical transformations

Response: We thank the referee for his/her careful reading!

4. Methods section, before equation (18): What is “ouch!” supposed to mean?

Response: We are sorry for our carelessness. We removed that “ouch”.

Reviewer 3

This is a very interesting paper on connections amongst several seemingly distinct models of ordering and the breaking of continuous rotational symmetry in stationary states far from thermal equilibrium. The authors show that the equal-time statistical properties of the ordered phase of an incompressible flock in two dimensions are related to the height fluctuation statistics of the KPZ equation in 1 + 1 dimension, which are known to be related to 2D smectics, and are shown further to be connected to a divergence-free XY model.

The paper is conceptually interesting, and a real understanding of the work places considerable technical demand on the reader. But this is because it is a very nontrivial piece of physics. Moreover, this is not arid theory: it makes sharp predictions for experiment. The main results and the nice pictorial interpretation of some parts of the calculation should appeal to the very broad readership that Nat Comm would like to address. I recommend acceptance once my questions are answered satisfactorily.

Response: We are glad that the referee found our paper interesting and thank him/her for recommending publishing our paper in Nature Communication.

The authors consider hydrodynamic flocking models in which the bulk modulus for the isotropic part of the pressure in the equation for the velocity vector field (the order parameter for this problem) goes to infinity, so that the velocity is solenoidal. A curious technical consequence is that the analysis even in the ordered phase requires keeping track of the components of the velocity along as well as transverse to the ordering direction participate significantly in the dynamics of the ordered phase. The dominant nonlinearity, moreover, turns out to be that governing the shape of the potential that controls the order-parameter magnitude, not the “advective” term. Both these results are in marked contrast to the situation prevailing in compressible flocking models. This is genuinely new physics introduced by the long-ranged constraint of incompressibility. The authors work out the consequences of these nonlinearities, ultimately via the connection to 1+1-dimensional KPZ.

Response: We are impressed by the referee’s keen eye spotting the key difference between the incompressible flocking model and the compressible one.

I have two remarks to which I would like the authors to respond.

1) The ordered phase of colloidal rollers studied by Bricard et al., ref [26] of the paper under review, would appear to be a realization of an incompressible 2D flock. If so, how does the flock of the present work escape the bend instability of the Bricard work (which they mention only in Supplementary Information, after eq. S42)?

Response: We thank the referee for pointing out to us this potential source of confusion. Bricard et al actually studied a two component fluid, with a compressible active component (the “Quinke” rotators) coupled to an incompressible passive one (the background fluid). This leads to a number of differences with our model, including the bending instability the referee mentions. We have clarified this in the introduction.

2) The authors say that the large-distance decay to zero of the connected part of the equal-time velocity correlator shows that long-range order prevails. As far as I can tell, it only shows that the full correlator $\langle v(r) \cdot v(r') \rangle$ approaches the disconnected part $\langle v \rangle \cdot \langle v \rangle$. It doesn’t establish that the latter is nonzero. Am I missing something?

Response: Please see the response to referee 2’s first minor comments/suggestions.

Reviewer 1

The authors have addressed all my comments appropriately, and accordingly have improved the manuscript. I therefore recommend for publication.

Response: We are glad that the referee recommends publication of our paper in Nature Communications.

Reviewer 2

In their revised version and response letter, the authors have satisfactorily addressed my questions and comments.

Response: We are glad that the referee found our response satisfactory.

The only minor point I remain unconvinced of is that Eqs. (2) and (3) are indeed necessary. It seems to me the main purpose of these two equations is to advertise future work not included in the present manuscript. I believe that it would be completely sufficient for the present paper to start from Eqs. (5), (6) and (4) and a short statement that the authors focus on the non-jammed constant density limit, in which all coefficients depend on the constant density and pressure is reduced to a Lagrange-multiplier for the incompressibility constraint (5). But this is a minor point and I am happy to recommend the revised version for publication in Nature Communications.

Response: We thank the referee for recommending publication of our paper in Nature Communications despite having reservations about this “minor” point. We have decided to take advantage of his/her indulgence and stick to our previous decision. We have done so because we believe that one of our surprising and important results is that there is more than one way to take the “incompressible” limit in active systems, in contrast to equilibrium fluids, for which this limit is unique. Furthermore, the referee’s suggestion that we state that we focus on the non-jammed limit might be quite confusing to readers if we have not first provided a context for it by discussing the other ways one can take the incompressible limit. For both of the reasons, we prefer to leave our discussion as it is.

Reviewer 3

I am satisfied with the authors’ response to my two remarks. I have also gone over their response to the points raised by the other referees and find their answers to be adequate.

Response: We are glad that the referee found our response satisfactory and adequate.

*A small quibble: ref. [17] shows theoretically that spontaneous uniaxial order in (apolar, the title notwithstanding) active systems is *unstable* when the hydrodynamic interaction is included. I don’t think a reference to this paper belongs where it’s cited, in the context of realizations of polar active systems that could show long-range order.*

Response: We thank the referee for his/her carefulness. We agree with the referee and have removed this reference from our paper.

Apart from this the paper can be published in its present form.

Response: We thank the referee for approving the publication of our paper in Nature Communications.